# Mechanistic and Therapeutic Insights into Ataxic Disorders with Pentanucleotide Expansions

**DOI:** 10.3390/cells11091567

**Published:** 2022-05-06

**Authors:** Nan Zhang, Tetsuo Ashizawa

**Affiliations:** Neuroscience Research Program, Department of Neurology, Houston Methodist Research Institute, Weil Cornell Medical College, Houston, TX 77030, USA; tashizawa@houstonmethodist.org

**Keywords:** neurodegeneration, microsatellite expansion diseases, pentanucleotide repeats, spinocerebellar ataxia, SCA10, SCA31, SCA37, CANVAS, RNA foci, TDP-43

## Abstract

Pentanucleotide expansion diseases constitute a special class of neurodegeneration. The repeat expansions occur in non-coding regions, have likely arisen from *Alu* elements, and often result in autosomal dominant or recessive phenotypes with underlying cerebellar neuropathology. When transcribed (potentially bidirectionally), the expanded RNA forms complex secondary and tertiary structures that can give rise to RNA-mediated toxicity, including protein sequestration, pentapeptide synthesis, and mRNA dysregulation. Since several of these diseases have recently been discovered, our understanding of their pathological mechanisms is limited, and their therapeutic interventions underexplored. This review aims to highlight new in vitro and in vivo insights into these incurable diseases.

## 1. Introduction

Microsatellite repeats of 2–6 nucleotides confer evolvability and play important roles in gene function, disease pathology, circadian rhythmicity, and variation in brain and behavioral traits [1,2]. With growing interest and improved diagnostics, the total number of microsatellite expansion diseases has reached 47. Thirty-seven of these diseases exhibit neurological or neuromuscular pathology, while the rest present with developmental defects [3]. The causative repeats can expand in both coding and non-coding regions, often leading to protein gain-of-function (GOF) and RNA GOF, respectively [4]. When expanded in the coding region, as frequently seen in polyglutamine (polyQ) diseases, the mutant protein changes its phase separation property to form nuclear and/or cytoplasmic aggregates [5,6]. On a molecular level, the mutant protein may show an altered interactome compared to the wildtype counterpart [7,8]. Many polyQ proteins are intrinsically involved in transcription regulation and DNA damage repair [9,10]. A pathological change in their biophysical and biochemical characteristics can impact neuronal survival. In addition, non-specifically targeting both alleles of the polyQ protein may interfere with normal cell functions, which could undermine the intended therapeutic efficacy and should warrant caution. When expanded in the non-coding region, the mutant RNA can incur GOF in forms of RNA foci formation, protein sequestration, spliceopathy, and repeat-associated non-AUG (RAN) translation [4,6,11,12,13]. Several C/G-rich repeat expansions in the 5′-untranslated regions (5′-UTRs) can alternatively cause protein loss-of-function (LOF) by CpG island and repeat hypermethylation and transcription silencing [14]. Many expanded repeat RNAs alone can form hydrogel or irreversible foci in cells [15]. This raises the question as to whether targeting repeat-flanking RNA sequences is sufficient to achieve foci clearance, albeit with improved specificity. The above-described RNA GOF mechanisms are largely applicable to pentanucleotide expansion diseases. This review will focus on four pentanucleotide expansion diseases that cause cerebellar degeneration (Table 1): spinocerebellar ataxia types 10/31/37 (SCA10/31/37) and cerebellar ataxia, neuropathy, and vestibular areflexia syndrome (CANVAS).

## 2. SCA10

### 2.1. Patient Demographics, Clinical Presentation, and Neuropathology

SCA10, an autosomal dominant neurodegenerative disease, is caused by ATTCT repeat expansion in intron 9 of the *ATXN10* gene. The core clinical phenotype is cerebellar ataxia with variably associated epileptic seizure [16]. Although the first cases of SCA10 were reported in families of Mexican origin [17,18,19], the disease was later discovered in subjects with Amerindian (Sioux native American) [20], Latin American (Brazil, Peru, and Bolivia) [21,22,23,24,25], and East Asian ancestries (China and Japan) [26,27]. SCA10 patients with Amerindian and Spanish admixture generally present ataxia and epileptic seizure, while those with Portuguese admixture mostly exhibit only ataxia [28]. Clinically, gait ataxia begins to manifest between ages 14 and 45. Other symptoms, such as limb dysmetria, dysdiadochokinesis, dysarthria, dysphagia, ocular abnormalities, cognitive impairment, or mood changes, often ensue. SCA10 patients with generalized motor seizures and/or complex partial seizures often display ATTCC repeats, or a combination of ATCCT and ATCCC repeat interruptions in the “pure” ATTCT tract [29,30,31]. This type of phenotypic exacerbation by repeat interruptions is the polar opposite to reduced penetrance by interruptions in other neurodegenerative diseases (such as Huntington’s disease [HD] and myotonic dystrophy type 1 [DM1]) [32,33,34]. Upon neurological examinations of four Mexican SCA10 families with seizure, abnormal electroencephalography (EEG) with diffuse cerebral dysfunction (±focal irritability) and cerebellar atrophy (sometimes accompanied by mild cerebral cortical atrophy) were observed, suggesting that the pathological process of SCA10 may involve multiple regions of the brain [35]. Extensive brain degeneration in the anterior cerebellum (lobules I–VI) was also observed in the later stages of SCA10 by a magnetic resonance imaging (MRI) study [36]. In contrast to the above observations, immunohistological examinations on postmortem SCA10 brains reported no apparent changes in the cerebral cortex, hippocampus, midbrain, and pons [37]. Atrophy occurs symmetrically in both cerebellar hemispheres, primarily affecting Purkinje cells (cell number, dendrite count, and arborization) and the molecular layer volume.

### 2.2. SCA10 Repeat Origin and Structures

SCA10 patients generally display profound intergenerational repeat instability and somatic mosaicism, which could confound the correlation between repeat size and age of onset in different studies [18,35,38]. The normal *ATXN10* allele often carries 10–31 ATTCT repeats, while the pathologically expanded allele could harbor 800–4500 repeats [39]. There is a high density of other repetitive elements in intron 9, including *Alu*, long interspersed element-1 (LINE-1) and long terminal repeat (LTR), representing approximately 32% of the total intron length [28,40]. During evolution, a LINE-1 was postulated to first insert into the *ATXN10* intron 9, followed by an *Alu* insertion in the middle of the LINE-1. The ATTCT repeat is located at the poly(A) tail of the *Alu* and could emerge from either a series of mutations of a TTTTT motif (in the reverse direction of the *Alu* poly(A) tail, from TTTTT to TTTCT and to ATTCT) or a direct conversion from an ancestral ATTTT motif [40].

Since the SCA10 repeat is 80% A/T-rich, the duplex DNA is expected to be thermodynamically unstable. Its structural properties were subsequently investigated under torsional stress in the context of plasmid. The threshold number of ATTCT repeats that preferentially unpairs under physiological superhelical density is about 8 [41]. Longer ATTCT repeats, (ATTCT)_29_, formed denatured bubbles under moderate superhelicities in atomic force microscopy (AFM). These denatured bubbles collapsed at high superhelicities and resulted in single-stranded, locally condensed structures that were still accessible to small chemicals and oligonucleotide hybridization. The above unusual observations have led to the hypothesis that ATTCT repeats may function as a DNA unwinding element (DUE) and contribute to repeat expansion and contraction through aberrant DNA replication initiation (see discussion below). When mixing (ATTCT)_9_ and (AGAAT)_9_ oligonucleotides in circular dichroism (CD), the resultant DNA duplex at 10 °C (annealing temperature) typically assumed a normal B-form without any unusual propensity to unpair [42]. However, when the annealing temperature dropped to 0 °C, the (ATTCT)_9_ strand formed a hairpin while the (AGAAT)_9_ strand remained unstructured. Formation of the (ATTCT)_9_ hairpin falls in line with the AFM observation and may impede complementary strands from reannealing. Recent high-resolution nuclear magnetic resonance (NMR) studies using (ATTCT)_2–5_ duplexes showed that the first two repeats folded into mini-dumbbells comprised of a regular TTCTA and a quasi TTCT/A pentaloops (Figure 1A). These mini-dumbbells may have functional importance in enhancing repeat instability (see discussion below) [43].

Many RNA repeats form unusual secondary structures that have a wide range of biological or pathological effects, such as CAG and CUG hairpins in polyglutamine diseases and DM1, respectively [44,45], and GGGGCC G-quadruplex in *C9orf72*-ALS/FTD (amyotrophic lateral sclerosis/frontotemporal dementia) [46]. An NMR study of an (AUUCU)_9_ oligonucleotide revealed an antiparallel A-form RNA hairpin composed of four ^5′^UAUU^3′^/^3′^UUAU^5′^ stems, each containing two A-U base pairs flanked by two U-U base pairs (Figure 1B); individual stems were separated by a C-C mismatch [42]. A later crystallographic study suggested that AUUCU repeats could exist as ^5′^UCU^3′^/^3′^UCU^5′^ loops enclosed by AU base pairs (Figure 1C, [47]). Nonetheless, large AUUCU repeats could potentially form complex secondary and tertiary structures with stability comparable to that of G/C-rich microsatellite repeats. Additional AUUCC, AUCCU, and AUCCC interruptions may further stabilize the metastable AUUCU 3D structure and contribute to interruption-specific pathology. Furthermore, the structured SCA10 repeat RNA may mimic internal ribosome entry sites (IRESs) to recruit the preinitiation complex in a cap-independent manner and to jump-start RAN translation. IRESs generally lack a common motif but are rich in stem loops and pseudoknots [48]. They become competitive when cap-dependent translation is repressed under stress conditions [49]. SCA10 repeat expansions may feed-forward RAN translation by inducing cellular stress (such as stress granule formation with cytoplasmic repeat foci) and by altering start codon fidelity via eIF2α phosphorylation, pathologically reminiscent of GGGGCC RNA [50]. Evidence to support such notions remains to be confirmed in SCA10 patient-derived cells and animal models.

### 2.3. RNA Gain-of-Function in SCA10 Cells and Animal Models

RNA-mediated toxicity plays a predominant role in disease pathology when repeat expansions occur in non-coding regions [4]. A toxic RNA gain-of-function has been proposed as the underlying patho-mechanism in SCA10. The *ATXN10* mRNA is strongly expressed in the human brain, heart, skeletal muscle, kidney, and liver as well as widely in juvenile and adult mouse brains [18,51]. Large SCA10 repeat expansions do not seem to impede transcription of *ATXN10* or its neighboring genes, at least not in the tested somatic cell lines [51]. During RNA processing, the AUUCU-containing intron 9 splices normally and shows no intron retention (which could affect nuclear retention/nucleocytoplasmic transport/cytoplasmic turnover) [52]. The fully processed *ATXN10* mRNA was detected at similar levels in normal and SCA10 patient-derived lymphoblasts, fibroblasts, and myoblasts as well as in human–mouse somatic cell hybrids [51]. Consistently, a 50% reduction in the *ATXN10* gene dosage did not cause the SCA10 phenotype in mice, suggesting that ATXN10 haploinsufficiency might not be a key patho-mechanism in SCA10 [51]. Completely in line with the mouse data, individuals carrying a familial translocation t(2,22) (p25.2; q13.31) that disrupts one *ATXN10* copy do not show any sign of ataxia or epilepsy [53].

The Ashizawa lab developed the RNA GOF hypothesis with the following observations [54]: (1) the spliced, expanded AUUCU repeats formed both nuclear and cytoplasmic foci in patient-derived fibroblasts, neuroblastoma cells ectopically expressing expanded AUUCU repeats, and 6-month-old transgenic SCA10 mouse brain; (2) the expanded AUUCU RNA induced apoptosis in neuroblastoma cells; (3) the AUUCU RNA pulled down heterogenous nuclear ribonucleoprotein K (hnRNP K) from mouse brain lysates and colocalized with hnRNP K in 6-month-old transgenic SCA10 mouse cortex; and (4) inactivation of hnRNP K triggered apoptosis and sequestration of hnRNP K by AUUCU foci led to protein kinase C delta (PKCδ) translocation to mitochondria and caspase-3 activation. This study generated the first transgenic SCA10 mouse model (Eno2-β-globin intron-LacZ in C57BL/6 background), where an interrupted 500-repeat tract was inserted in a rabbit *globin* intron located upstream of the *LacZ* reporter. The transgene expression was driven by the rat neuronal *enolase* (*Eno2*) promoter. Given that these mice breed poorly and that apoptosis is not detected in the brain [55], a second transgenic SCA10 mouse line was developed (Prnp-LacZ-3′UTR in FVB/N background). In this model, an interrupted 500-repeat tract was inserted in the 3′-UTR of *LacZ* and its expression was driven by the rat prion (*Prnp*) promoter [56]. Histopathologically, neurodegeneration was predominantly confined to the hippocampal CA3 region in 6-month-old mice, but neither neuronal loss, apoptosis, nor gliosis was observed in the cerebellum—the site of atrophy in SCA10 patients. Both cytoplasmic and nuclear AUUCU foci were readily detected in 6-month-old transgenic cortexes, pontine nuclei, and hippocampal neurons, with concomitant hnRNP K sequestration and PKCδ translocation in many parts of the brain. Phenotypically, this transgenic SCA10 line displays locomotive dysfunction and seizure behavior.

hnRNP K is an RNA/DNA-binding protein involved in a wide range of regulatory processes [57]. It has three RNA/DNA-binding homology domains (KH1–3) [58]. All KH domains structurally elucidated to date are highly conserved; they bind to single-stranded RNA by forming favorable interactions with at least four nucleotides (N1–N4) [59]. The nucleotide preference for each position is: U/C/A for N1, C for N2, A/C for N3, and C for N4. hnRNP K may recognize the canonical SCA10 repeat via its KH domains in two possible registers (AU^1^U^2^C^3^U^4^ or A^1^U^2^U^3^C^4^U), whereas repeat interruptions (such as AU^1^C^2^C^3^C^4^) may better conform to the consensus and further enhance hnRNP K binding and sequestration. It should be noted that hnRNP K is also a potent tumor suppressor, as the haploinsufficiency of hnRNP K is a driver of acute myeloid leukemia (AML) [60]. Given that hnRNP K is sequestered by AUUCU RNA foci, it is reasonable to speculate a haploinsufficiency scenario in SCA10 where the functional hnRNP K pool is drastically reduced or depleted, leading to an increased risk of cancer. A recent clinical study, however, showed no significant correlation between SCA10 and presence of cancer [61]. hnRNP K can also be fittingly described as an oncogene, since its overexpression correlates with poor prognosis and advanced disease status in a variety of malignancies [60]. It thus warrants caution to therapeutically modulate hnRNP K expression in SCA10.

### 2.4. Proposed Models of SCA10 Repeat Expansion

Three models have been developed to explain the molecular mechanisms of SCA10 repeat instability: the aberrant replication re-initiation model and the template-switching model accounting for large-scale ATTCT expansions [41,62], and the proof-reading escape model by mini-dumbbells accounting for expansions two repeats at a time [43]. (1) In the replication re-initiation model, longer ATTCT repeats (ATTCT_23/27/48_) function as a DNA unwinding element (ATXN10 DUE) that supports plasmid replication in a HeLa cell extract and substitutes for the *c-myc* DUE in human cells [41,63]. Cells that contain longer ATTCT repeats and have undergone 250 population doublings show a dramatic increase in repeat length [63]. These data point to an aberrant replication model where re-initiation occurs at multiple sites within the ATXN10 DUE, which may worsen with increasing repeat lengths. The branched DNA molecule (Figure 2A top panel) or the “onion skin” configuration (Figure 2A bottom panel) could result in recombination, strand break, and repeat incorporation into the nascent strand, leading to large-scale expansions. (2) The template-switching model was developed based on studies in yeast. In contrast to human data [51], ATTCT expansions (ATTCT_46/64/81_) significantly reduced transcription both upstream and downstream of the repeat and signaled premature polyadenylation [62]. When the *TOF1* gene (encoding a fork-stabilizing protein) was deleted from the yeast strains carrying (ATTCT)_64 or 105_, both repeat expansion and contraction rates were increased by 5-fold and 7.5-fold, respectively. The *RAD5* (encoding a template-switching protein)-knockout in the same reporter strains resulted in complete elimination of repeat expansion but left the contraction rate largely unchanged. The above observations have led to the hypothesis that template-switching is key to repeat expansion, where the nascent leading strand may hybridize to an Okazaki fragment. Upon reaching the end of the Okazaki fragment, the leading strand may switch back to the template strand, thus incorporating additional repeats and allowing the replication fork to advance (Figure 2B). (3) The proof-reading escape model was based on in vitro primer extension experiments [43]. When an (ATTCT)_2_ mini-dumbbell is located ≥5 nucleotides from the 3′-end of the nascent strand, it can escape the proof-reading activity of DNA polymerase (Figure 2C), thus feeding two repeats into the expansion at a time. However, it is not known how these mini-dumbbells escape from mismatch repair that can detect internal loops up to 17-nucleotide long and why they do not stall DNA polymerase.

### 2.5. Theranostic Development for SCA10

SCA10 can often manifest ataxia without epileptic seizure; thus, its diagnosis should be distinguished from other SCAs by accurate genetic screening. Southern blot has been the golden standard for determining various microsatellite expansions. However, this requires a large amount of high-quality patient DNA (5–10 µg) that could be difficult to procure. Next generation sequencing technologies, such as single molecule real-time (SMRT) and nanopore, have made giant leaps from their predecessors in reading long repetitive sequences with incredible accuracy. However, these technologies have not yet reached economy of scale. We have recently developed a FEMTO-pulse-based assay in conjunction with repeat-primed PCR [64]. By testing on as little as 50 pg of PCR products from SCA10 patient DNA or SCA10 yeast artificial chromosome, not only could the repeat range be determined in a cost-effective manner, but also the unbiased heterogeneity within the same sample.

To target AUUCU repeat RNA in SCA10, the Disney group screened a library of small molecules and found a bis-benzamidine compound 2 (2AU) that binds to AU base pairs with high affinity and specificity [65]; AU base pairs flank the ^5′^UCU^3′^/^3′^UCU^5′^ loops in the SCA10 repeat crystal structure (Figure 1C). This lead compound was further optimized to display two 2AU modules (valency of 2) separated by two propylamine spacer modules (final compound denoted as 2AU-2). AUUCU_500_ was pulled down by the dimeric 2AU-2 from total cellular RNA, and its enrichment was diminished when a 100-fold excess of AUUCU_11_ was used for competition [65]. The preferential binding of 2AU-2 to expanded repeats is faithfully mirrored by another dimeric module designed to target long CUG repeats [66]. Interestingly, a fraction of small RNA and tRNA were simultaneously enriched with AUUCU_500_, but the binding of 2AU-2 to tRNA at active concentrations did not interfere with normal cellular translation. Further in vitro studies suggest that 2AU-2 could significantly reduce foci count, caspase-3 activation, and PKCδ translocation in SCA10 patient-derived fibroblasts [65].

## 3. SCA31

### 3.1. Patient Demographics and Neuropathological Features

SCA31 is regarded as a “pure” cerebellar ataxia with autosomal dominance caused by TGGAA repeat expansion. It is ranked the third most frequent SCA in Japan (after SCA3 and SCA6), very rarely found in neighboring Asian countries [67,68,69,70], and not found in large European SCA cohorts [71]. SCA31 has also been diagnosed in Brazilian SCA patients with Japanese ancestry, suggesting a strong founder effect [72]. The age of onset for SCA31 is 58.5 ± 10.3 years, making it the latest to manifest of all SCAs [73]. Patients typically become wheelchair-bound at 79.4 ± 1.7 years and decease at 88.5 ± 0.7 years. Clinical features of SCA31 include truncal and limb ataxia, cerebellar speech, and reduced muscle tonus [74]. Some patients additionally develop nigrostriatal dopaminergic dysfunction, L-DOPA responsive parkinsonism, and blepharospasm followed by ataxic dysarthria [75,76]. Histologically, the cerebrum, brainstem, and spinal cord all appear normal in SCA31 patients, while the cerebellar cortex shows signs of degeneration [74,77,78]. In the affected regions, Purkinje cells not only drop in number, but also show shrinkage and halo-like amorphous materials surrounding the cells. These amorphous materials are composed of calbindin-positive somatic sprouts (cactus-like formation) and synaptophysin-positive presynaptic terminals innervated from basket cells, neurons of the inferior olivary nucleus, or other neurons that connect to Purkinje cells [77,79]. Ubiquitin-positive degradation granules and Golgi fragmentation are also enriched in the amorphous materials [77,79].

### 3.2. RNA Gain-of-Function and Pentapeptide Repeat Protein in SCA31

SCA31 is caused by a 2.5- to 3.8-kilobase insertion containing (TGGAA)_n_, (TAAAA)_n_, (TAGAA)_n_ and (TAAAATAGAA)_n_ in an intron shared by two genes: *brain expressed associated with NEDD1 (BEAN1)* and *thymidine kinase 2 (TK2)* on chromosome 16q22.1—a locus previously known for but now shown to be distinct from SCA4 [80] (Figure 3A). The majority of healthy controls (99.77%) do not have any insertions or, on very rare occasions (0.23%), have an insertion without the TGGAA repeats. It has therefore been proposed that TGGAA repeats are causative to SCA31, and it has been shown that the insertion length inversely correlates with age of onset. Interestingly, a single nucleotide change (AB473217) also co-segregates with the disease [80]; it may have an unknown regulatory role over the repeats (such as a non-coding RNA).

Haploinsufficiency may not be a major contributor to SCA31 pathology, since the mRNA expression and splicing levels of *BEAN1*, *TK2*, or other nearby genes are similar in control and patient cerebella [80]. It should be noted that the above observations are largely based on cell-type averaging, given that Purkinje cells are the only affected neurons in SCA31 and that they represent a small fraction of total cerebellar mass. An RNA GOF was subsequently proposed for SCA31 when several fluorescent in situ hybridization (FISH) assays detected RNA foci in approximately 30% of patient Purkinje cells, using probes targeting either (UGGAA)_n_ or (UAAAAUAGAA)_n_ repeats (Figure 3A) [78,80]. These SCA31 RNA foci were only detected in the *BEAN1*-direction (no anti-sense foci), ranged from 0.2 to 1.8 µm in diameter, and accumulated in the nuclei of Purkinje cells (not observed in other neurons or glial cells) [78,80]. When the TGGAA expansion was expressed in PC12 cells, cell toxicity and death were significantly elevated [78]. When the expression of (TGGAA)_80–100_ repeats was induced under the *GMR-GAL4* driver in *Drosophila*, extensive nuclear and cytoplasmic foci in eye imaginal discs and severe eye degeneration were observed [81]. Phenotypically, when the TGGAA expansion was induced in the nervous system under the *ELAV*-GeneSwitch driver, flies exhibited a shorter lifespan and progressive locomotive defects with age [81].

Like any “pure” pentanucleotide expansion, translation of UGGAA repeats in any reading frame would yield a single pentapeptide repeat (PPR) protein, namely poly-WNGME in SCA31 (Figure 3A). In SCA31 patient brains, two separate anti-PPR antibodies detected granular structures in Purkinje cell body and dendrites, but not in control brains [81]. In parallel, PPR expression was detected in the cytoplasm of cells in eye imaginal discs of SCA31 transgenic flies but not in non-transgenic or non-expanded flies, and the PPR protein level correlated positively with the severity of eye degeneration [81]. The above observations suggest that the aberrant repeat translation contributes to SCA31 pathology and that RNA foci, in addition to its toxicity, and may function as an RNA sink to protect against PPR formation in the cytoplasm. Given that the SCA31 UGGAA repeats are flanked by two UAGAA repeats (UAGAAUAGAAUGGAA, Figure 3A), it is currently unknown whether the PPR protein arises from AUG-dependent translation or RAN translation.

### 3.3. Interplay between SCA31 RNA Expansion and RNA Binding Proteins

The heat shock response (HSR) is a highly conserved process that is tightly regulated by the master heat shock transcription factor 1 (HSF1). Upon thermal stress, HSF1 activates the transcription of long arrays of Satellite III (SatIII) in pericentromeric heterochromatic regions. The SatIII RNA is rich in GGAAT repeats (GGAATGGAAT as in SCA31 [82]), remains associated with the site of transcription [83], and recruits several RNA binding proteins (RBPs) to form nuclear stress bodies (nSBs) for global transcription repression (Figure 3B) [84]. Amongst the recruited RBPs are serine/arginine-rich splicing factors 1 and 9 (SRSF1 and SRSF9) [85,86]—both bind to SCA31 repeats in vitro as they do SatIII [80]. It is possible that the SCA31 UGGAA repeat competitively inhibits SatIII function [87], or it mimics nSBs to induce global transcription repression. Transcription dysregulation has been observed in several neurodegenerative diseases (such as HD and SCA3 [88,89]) but remains to be reported in SCA31.

Several ALS/FTD-related RNA binding proteins, such as TAR DNA binding protein 43 (TDP-43), FUS and hnRNP A2B1, were identified by unbiased UGGAA RNA pulldown and mass spectroscopy using mouse brain lysates (Figure 3B) [81]. TDP-43 colocalizes with SCA31 foci in patient Purkinje cells and its affinity for UGGAA RNA increases with repeat size. When TDP-43 was co-expressed in SCA31 transgenic fly eyes, it restored pigmentation and ommatidial formation as well as suppressed RNA foci formation [81]. TDP-43 may function as an RNA chaperone owing to: (1) its inability to affect UGGAA RNA degradation, and (2) its ability to bind and structurally alter UGGAA RNA, as demonstrated by CD and AFM. TDP-43 also suppressed PPR protein synthesis in SCA31 transgenic flies. The above rescue effects by TDP-43 were closely mirrored by two other UGGAA-binding proteins—FUS and hnRNP A2B1. Interestingly, the co-expression of non-expanded (UGGAA)_22_ in ALS flies (expressing TDP-43 G298S, FUS, or hnRNP A2B1 D290V) rescued eye degeneration and protein aggregation [81,87]. This result is consistent with previous findings where the binding of UG repeat RNA or Clip_34nt RNA to TDP-43 inhibits protein aggregation in vitro [90,91]. It is likely that an optimal RNA-to-protein ratio helps maintain the reversibility of RBP phase separation and protects the protein from undertaking hydrogel or insoluble amyloid plaque formation [87]. Taken together, the above data stress the importance of maintaining a functional equilibrium between RNA and RBP; any imbalance on either side of the equilibrium (by RNA expansion or proteinopathy) may lead to neurodegeneration [81,87].

### 3.4. Small Molecule Targeting UGGAA in SCA31

The UGGAA RNA has been proposed to fold into a hairpin with ^5′^GGA^3′^/^3′^AGG^5′^ loops flanked by AU base pairs [92]. A small molecule screen recently identified a naphthyridine carbamate dimer (NCD) that preferentially binds to UGGAA repeats. The stoichiometry between NCD and the UGGAA/UGGAA pentad is 2:1, as demonstrated by mass spectrometry and NMR (Figure 3C), where four naphthyridine-guanine base pairs stack between two A-U base pairs [92]. NCD not only suppresses SCA31 RNA foci and heat shock-induced nSB formation in HeLa cells, but also reduces the interaction between UGGAA and TDP-43, SRSF9, and hnRNP M. More importantly, when NCD was fed to SCA31 transgenic larvae, the compound eye degeneration in adult flies was ameliorated [92]. The above data suggest that targeting the G-rich mismatched base pairs by NCD may be a valuable therapeutic strategy to test in animal models.

## 4. SCA37

### 4.1. Clinical Presentation and Neuropathology

SCA37 is an autosomal dominant neurological disorder that has been identified in Spanish and Portuguese kindreds [93,94]. The causative mutation is mapped to an ATTTC expansion in a 5′-UTR intron of *DAB1* (*disabled 1*) [93]. The mean age of onset is 43.3 ± 9.9 years, with initial clinical presentations including dysarthria, falls, and vertical eye movement abnormalities. The disease progressively evolves to a “pure” ataxia with scanning speech, dysphagia, mild truncal ataxia, and severe dysmetria mostly in the legs [95]. Under postmortem examinations, the SCA37 patient cerebral hemispheres, brainstem, and spinal cord were unremarkable, while the cerebellar cortex showed an extensive and generalized Purkinje cell loss, accompanied by dendritic and nuclear deformity and ubiquitin-positive perisomatic granules immunostained for DAB1 [95]. The molecular layer exhibited diffuse astrogliosis and a significant reduction in thickness.

### 4.2. RNA Toxicity and Dysregulation of DAB1 Expression in SCA37

By comparing DNA from a large Portuguese kindred to the reference genome, the Silveira group revealed that the *DAB1* pathogenic allele contains (ATTTT)_60–79_(ATTTC)_31–75_(ATTTT)_58–90_, while the normal allele harbors (ATTTT)_7–400_ repeats [93]. An inverse correlation exists between ATTTC repeat length and age of onset in SCA37, with a gender-specific contribution identified only in affected males [93,95]. When transiently transfected into HEK293T cells, the (AUUUC)_58_ RNA, but not the normal or expanded (AUUUU)_7 or 139_ RNA, formed nuclear foci [93]. When injected into one- or two-cell stage zebrafish embryos, the (AUUUC)_58_ RNA showed a higher lethality rate (58.79%) than the control RNAs (18.14% on average). The developmental rate of (AUUUC)_58_-injected embryos (7.76%) was significantly lower than that of control-injected embryos (80.89% on average). The above data highlight the toxicity of AUUUC RNA in vivo.

The *DAB1* gene encodes an adaptor protein in the reelin signaling pathway that is responsible for accurate laminar migration of neurons during brain development [96]. *Dab1*-deficient mice (scrambler or yotari) all lack cerebellar foliation and display ectopic Purkinje cell clusters deep in the cerebellar cortex [97,98,99]. To date, eleven *DAB1* mRNA species have been identified in humans, seven of which code for proteins (Ensembl). The cap analysis of gene expression (CAGE) on four *DAB1* mRNA species—arising from alternative non-coding exon usage—showed that their expression was largely brain specific, with higher levels in human fetal brain tissue than in adult brain tissue [93]. By analyzing *DAB1* mRNA isoforms from postmortem brains, the Matilla–Duenas group identified a *DAB1-4* transcript (containing coding exons 20 and 21) uniquely present in SCA37 but absent in control cerebella (Figure 4A). This observation was further corroborated by RNAseq mining in ENCODE [95]. Overexpression of these two evolutionary conserved coding exons in mice (*Dab1.7bc*) was sufficient to antagonize wildtype (WT) Dab1 function and induce neuronal migration defects [100]. Their exclusion is normally regulated by the neuronal splicing factor Nova2 that binds to the YCAY clusters upstream of exons 7b and 7c of *Dab1* in mice (Figure 4B). Restoration of the Dab1/Dab1.7bc ratio by overexpressing WT Dab1 rescues the neuronal migration defect in *Nova2* knockout mice [100]. Contrary to the mouse data, WT DAB1 (80 kD) is significantly elevated in both cerebellar vermis and hemisphere of SCA37 patient brain, which in turn over-activates PI3K and AKT along the reelin signaling pathway (Figure 4A). Given that tau (MAPT) is a downstream target of reelin/DAB1, one possible patho-mechanism of SCA37 could be tau-related microtubule defects. However, immunostaining for tau only revealed moderate neuritic plaques in SCA37 patient cerebellum and the cytoarchitectural and neuronal populations were well preserved in these regions [95]. Interestingly, in silico prediction suggests that ATTTC repeats form putative binding sites for the transcription factor XBP1 (X-box binding protein 1, Figure 4A) [95]. XBP1 is known to be activated during embryo stage and during ageing by the unfolded protein response, but downregulated in adulthood [101]. It is currently not known how XBP1 contributes to SCA31 pathology with ageing and how it feeds into the regulation of *DAB1* mRNA with respect to Nova2.

## 5. CANVAS

### 5.1. Clinical Presentation and Neuropathology

CANVAS is an adult-onset, autosomal recessive neurodegenerative disease with a spatial progression from the early sensory neuronopathy to the later appearance of bilateral vestibulopathy and cerebellar ataxia [102,103]. A causative biallelic AAGGG repeat expansion in intron 2 of the *replication factor C1* (*RFC1*) gene has been identified in tested familial and sporadic CANVAS cases [104,105]. This expansion is thought to have arisen 25,000 years ago in Europe and estimated with a biallelic carrier frequency of 1:400–1:10,000 at birth [102,105]. The core sensory neuronopathy is consistent with the thinning of the peripheral nerves, sensory neuron deprivation in the dorsal root ganglia (ganglionopathy), and posterior column atrophy of the spinal cord [106,107,108,109]. Some patients develop widespread demyelination in nerve biopsies as well as chronic denervation with reinnervation in muscle biopsies [102,104]. Cerebellar atrophy is present in the majority of CANVAS cases (83% in one study [104]); Purkinje cell loss occurs in both cerebellar vermis and hemisphere with empty baskets, torpedoes, and prominent Bergmann gliosis [104,106,107,110]. Additional symptoms, such as chronic cough (in 97% patients), oculomotor signs (85%), motor neuron involvement (63%), dysautonomia (50%), and Parkinsonism (10%), have been observed in clinically and genetically diagnosed CANVAS patients [110]. Examination of CANVAS postmortem brains revealed complex immunohistological patterns: (a) Cortese et al. demonstrated an age-related tau neurofibrillary tangle in the medial temporal lobe and amyloid-β deposits in the neocortex, subiculum, CA1, and caudate nucleus. There were no obvious inclusions of p62, α-synuclein, or TDP-43 in the brain. The brainstem and substantia nigra showed no apparent neuronal loss [104]. (2) Huin et al. demonstrated abundant α-synuclein Lewy bodies in the locus coeruleus, substantia nigra, hippocampus, amygdala, and entorhinal cortex, consistent with the patient’s Parkinsonism. Tau inclusions were detected in the entorhinal cortex and hippocampus, while amyloid-β immunoreactivity was absent in all tested brain regions. Marked astrogliosis and axonal swelling of motor neurons were observed in the anterior horn at the lumbar level without TDP-43 or p62 inclusions [110]. (3) Reyes-Leiva et al. did not detect cytoplasmic or nuclear inclusions of ubiquitin, p62, tau, α-synuclein, phosphorylated-TDP-43, FUS, or polyQ in the brainstem, spinal cord, or cerebellum. There was a moderate reduction in motor neurons in the anterior horn at the thoracic level, which was more than expected from ageing [109].

### 5.2. Repeat Polymorphism and Intron Retention for CANVAS

The *RFC1* repeat locus is incredibly polymorphic. In control populations, the allelic distribution is: 75.5% for (AAAAG)_11_, 13.0% for the expanded (AAAAG)_15–200_, 7.9% for the expanded (AAAGG)_40–1000_, 0.7% for the expanded (AAGGG)_400–2000_, and the remaining 3% for novel repeats (such as AAGAG or AGAGG) [104,111]. In contrast, the biallelic AAGGG expansion is detected in 92% of tested CANVAS cases and 100% of familial cases [104]. There is a positive correlation between repeat size and repeat G/C-content [112], but no correlation exists between repeat size and age of onset [102]. Recently, two new recessive repeat expansions have been linked to disease pathology: the (ACAGG)_n_ repeat in Asian Pacific and Japanese cohorts, often presenting fasciculations and elevated serum creatine kinase [112,113], and the (AAAGG)_10–25_(AAGGG)_n_(AAAGG)_n_ repeat in Maori [114].

The pathological mechanism of CANVAS may not be a classical RNA GOF or protein LOF in that [104]: (1) the *RFC1* mRNA levels are similar in control and patient cortexes, cerebella and peripheral cells (fibroblasts and lymphoblasts); (2) no sense or antisense RNA foci were detected in patient’s cerebellar vermis, as opposed to SH-SY5Y cells transfected with either (AAGGG)_54_ or (TTCCC)_94_ where nuclear and cytoplasmic foci were clearly visible; (3) RFC1 protein (isoform 1) was expressed at the same level in control and patient brains as well as in peripheral cells; and (4) even though RFC1 plays a key role in DNA replication and damage repair and is extremely intolerant to LOF [115], patient fibroblasts did not show increased susceptibility to double-stranded DNA breaks. Nonetheless, the assessment of *RFC1* pre-mRNA expression revealed a consistent increase in intron 2 retention in patients’ cerebellum, frontal cortex, lymphoblast, and muscle, when compared to healthy controls [104]. Intron retention has been selectively linked to G/C-rich microsatellite expansions and has a profound impact on nucleocytoplasmic transport and neuronal gene expression [52]. Its pathological contribution to the core CANVAS phenotypic triad is currently unknown and will require further investigation.

## 6. Conclusions and Future Directions

The pentanucleotide expansions described here all reside in the poly(A) tail of intronic *Alu*s in either sense or antisense direction [116]. The mobile and mutagenic nature of *Alu* may contribute to not only repeat diversity but also large intergenerational and somatic instability in patients. Given that these repeats are A/T-rich and share evolutionary ties, they may engage an overlapping group of RBPs with key functions in maintaining cerebellar neuronal health. The functional compromise of these RBPs may explain why the cerebellum is the primarily affected brain region, even though the expanded pentanucleotide RNA is ubiquitously expressed. Disruption of the delicate balance between RNA and RBPs by repeat expansion may lead to Purkinje and cerebellar neurodegeneration—a common theme amongst the pentanucleotide expansion diseases. Other disease-specific pathological features may arise from interrupting repeats within the “pure” tract. The interrupting repeats diverge from the canonical repeats by accumulating G/C-rich mutations. A high G/C-content has been shown to encourage repeat expansion [112]. They may also conform to new consensus motifs recognized by different RBPs and cause cellular defects that are formerly associated with G/C-rich tri- or hexa-nucleotide expansions. Many pentanucleotide repeats are flanked by multiple ATG start codons at their 5′-ends (Ensembl), and themselves contain imperfect start codons (such as ATTCT, ATTTC and AAGGG, Table 1). Although imperfect start codons initiate translation less efficiently, the presence of secondary structures downstream often markedly increases the efficiency [50]. The expanded pentanucleotide RNAs are known to form complex secondary structures. It is possible that they drive the toxic PPR protein synthesis by an amalgamation of AUG-dependent and non-AUG-dependent, IRES-like mechanisms. Given their intronic nature and normal splicing pattern, the expanded pentanucleotide RNAs should not possess a 5′-m7G cap that is usually acquired co-transcriptionally. Thus, PPR protein synthesis most likely occurs in a cap-independent manner.

The spliced RNAs exist in a lariat form and become debranched and degraded by nuclear exonucleases; they are usually not exported to the cytoplasm [117]. The expanded pentanucleotide RNAs may be resistant to processing and degradation and accumulate to form nuclear foci. Conversely, there must be nucleocytoplasmic transport mechanisms that actively export the expanded RNA into the cytosol. For instance, SRSF1 has been shown to bind GGGGCC RNA and mediates its cytoplasmic transport via NFX1 (nuclear transcription factor X box-binding protein 1) [118]. SRSF1 also binds to UGGAA RNA in SCA31 [80]. It is likely that key RBPs with degenerate recognition motifs are “hijacked” by multiple pentanucleotide repeat types for their cytoplasmic triage and subsequent PPR synthesis. It is therefore imperative to unbiasedly identify and cross-reference the protein partners of different repeat types. To date, large scale RNA sequencing (RNAseq) and unbiased protein pulldown experiments using animal brains and cell models of SCA31, SCA37 and CANVAS are still missing. We and our collaborators have performed RNAseq analyses on SCA10 patient and mouse brains but, surprisingly, did not find significant splicing alterations in comparison to WT samples (unpublished data).

It is currently unclear as to why A/T-rich repeat expansions cause an autosomal dominant inheritance in SCA10, SCA31, and SCA37, but an autosomal recessive inheritance in CANVAS. Given that no apparent changes in mRNA and/or protein expression of the expanded genes were observed between normal and patient samples (of all four discussed diseases), a protein LOF and its related gene dosing effect are unlikely a major contributor to the different inheritance patterns. This is particularly interesting for CANVAS, as Friedreich ataxia, another autosomal recessive neurodegenerative disease caused by A-rich repeat expansions, exhibits a marked reduction in mRNA and protein dosing of *frataxin* [119,120]. The causative gene of CANVAS—*RFC1*—plays an important role in DNA replication and damage repair. It is possible that compensatory mechanisms are in place in individuals with one mutant *RFC1* allele, while in individuals with two recessive alleles, ageing and the accumulation of DNA damage may eventually overrun the repair mechanism, leading to late-onset neurodegeneration. Indeed, certain mutations in DNA damage repair proteins have been linked to autosomal recessive forms of ataxia and Parkinson’s disease [121]. It is also interesting to note that no sense or antisense RNA foci were observed in CANVAS patient brain, but *RFC1* intron 2 retention is consistently elevated in multiple patient brain regions and peripheral tissues [104]. Intron retention typically occurs in G/C-rich, autosomal dominant expansion diseases (such as DM, Fuch’s endothelial corneal dystrophy, and C9orf72), but not in A/T-rich expansion diseases (such as Friedreich ataxia and SCA10); it has a wide cellular impact on tissue development, neuronal gene expression, RNA nuclear retention, and nucleocytoplasmic transport, to name but a few [52]. It is currently unknown as to how intron retention in CANVAS links to disease etiology and autosomal recessive inheritance. As a final note, pentanucleotide DNA or RNA repeats constitute a well-defined target upstream of the pathological cascade. Our therapeutic toolbox has expanded beyond small molecules to biological modalities, such as DNA- and RNA-editing CRISPRs [122,123,124,125,126,127,128], antisense oligonucleotides (ASOs) [129,130,131,132,133,134,135], and DNAzymes [136]. For instance, Cas9 proteins complexed with single- or dual-guide RNAs have been used to excise repeat expansions across multiple neurodegenerative diseases [122,123,124,128,137,138]. Their relatively large cassette size with respect to adeno-associated viral (AAV) packaging capacity, potential immunogenicity, and off- and on-target deletion and chromosomal rearrangement still pose concerns about the efficacy and safety of permanent DNA editing [122,139,140]. By using Cas-guide mRNAs or ribonucleoproteins encapsulated in lipid nanoparticles (LNPs) for transient expression [141], deactivated Cas9 (dCas9) for blocking repeat transcription [125,142], dCas9 conjugates for repeat RNA degradation [126], or compact Cas proteins (such as CasX, Cas14 or Casφ) [143,144,145], some of the above obstacles could be circumvented. Alternatively, RNA editing may offer a “safer” approach that would require frequent dosing. For instance, Cas13-based systems have been successfully tested for RNA elimination in DM1 myoblast [127], cancer cells [146,147], and coronavirus-infected cells [148]; it is currently unknown whether the collateral RNA cleavage by Cas13 would outweigh its therapeutic benefits. However, it is reassuring that a Cas13d-based glia-to-neuron conversion significantly alleviated retinal injury in vivo and motor defects in a Parkinson’s mouse model without inducing apparent toxicity [149]. In parallel, both ASO and DNAzyme are RNA-targeting single-stranded DNAs that are compatible with a wide range of chemical modifications; the difference in their modus operandi is that ASO requires cellular RNase H activity, while DNAzyme relies on its own catalytic function induced by ions and 3D folding [150]. Although chemically modified ASOs (and by inference DNAzymes) can already achieve wide central nervous distribution in rodents and non-human primates [151], their packaging with LNP and bio-conjugation with cell-penetrating peptides, receptor-targeting antibodies, or ligands may further facilitate the passage across the blood–brain barrier, cellular uptake, and endosomal escape. RNA interference may be more efficacious at clearing cytoplasmic foci and PPR-related toxicity due to the cytoplasmic compartmentalization of the effector complex. With these technical advancements and progress on viral and non-viral delivery strategies, new hopes can be offered to patients under a clear definition of disease mechanisms.

## Figures and Tables

**Figure 1 cells-11-01567-f001:**
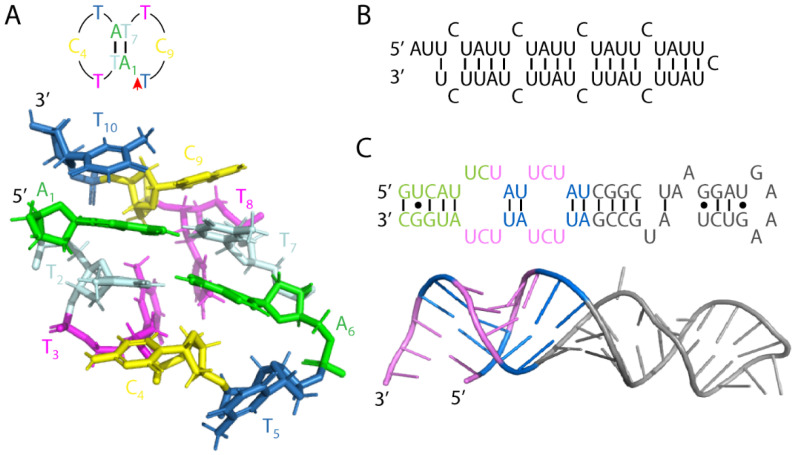
SCA10 repeat DNA and RNA structures. (**A**) Schematic and high-resolution NMR structure (PDB entry 6IY5) of an (ATTCT)_2_ DNA mini-dumbbell. Each mini-dumbbell is composed of a TTCTA and a quasi TTCT/A pentaloops. The red arrow indicates the backbone discontinuity site. (**B**) The proposed NMR structure of an (AUUCU)_9_ RNA hairpin. (**C**) Schematic and crystal structure (PDB entry 5BTM) of two (AUUCU)_2_ repeats separated by 25 ribonucleotides (grey). The ^5′^UCU^3′^/^3′^UCU^5′^ loops and AU base pairs are highlighted in pink and blue, respectively. Green ribonucleotides are missing in the final crystal structure.

**Figure 2 cells-11-01567-f002:**
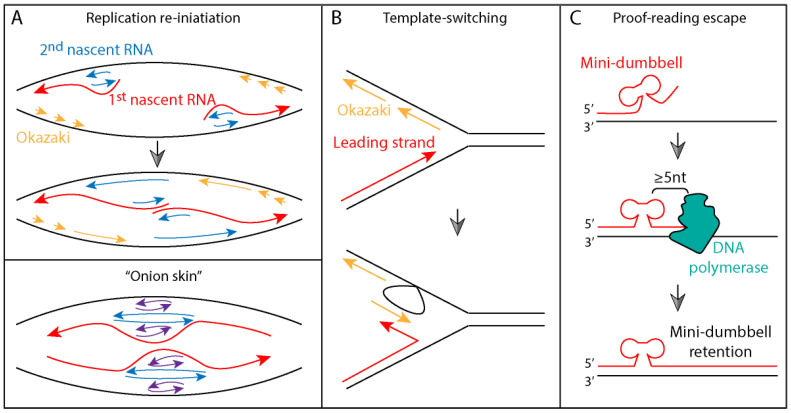
Three proposed models for SCA10 repeat expansion. (**A**) In the replication re-initiation model, multiple DNA unwinding events occur simultaneously at the ATTCT DUE. The branched DNA or “onion skin” layers, formed between nascent strands or between nascent and template strands, undergo recombination and end joining to give rise to large scale expansions. They may also undergo strand breakage associated with replication fork collapse to cause repeat contraction. (**B**) In the template switching model, the nascent leading strand may switch from its template to hybridize to an Okazaki fragment due to sequence complementarity. Upon reaching the end of the Okazaki fragment, the nascent leading strand may switch back to its original template. This will allow repeat to expand and replication fork to continue. (**C**) In the proof-reading escape model, if the mini-dumbbell formed by (ATTCT)_2_ is located ≥5 nucleotides (nt) away from the nascent 3′-end, it can escape from the proof-reading by DNA polymerase, thus adding two repeats into the expansion.

**Figure 3 cells-11-01567-f003:**
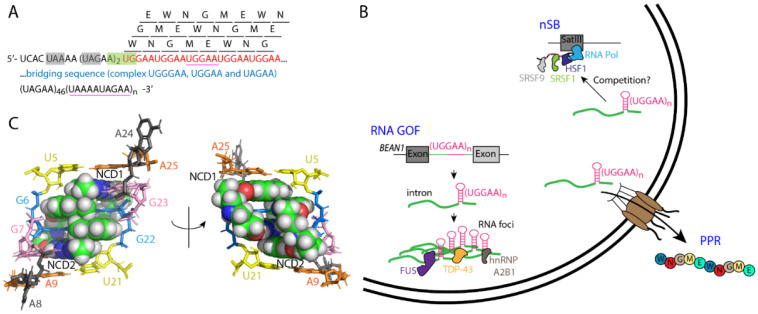
SCA31 RNA-mediated toxicity and PPR protein synthesis. (**A**) The insertion sequence with UGGAA repeat expansion (red) in SCA31 patients. The bridging sequence is highlighted in blue. Translation of UGGAA repeats in all 3 reading frames gives rise to a single PPR protein, poly-WNGME (single letters for amino acids). Stop (UAA and UAG) and start (AUG) codons are highlighted by grey and green boxes, respectively. The FISH probe target sites for RNA foci detection are underlined in purple. (**B**) RNA-mediated pathology in SCA31. The expanded UGGAA RNA could (1) form RNA foci that sequester TDP-43, FUS, and hnRNP A2B1; (2) compete with UGGAA-rich SatIII RNA that is induced by thermal stress to form nSB with SRSF1, SRSF9, and HSF1; and (3) be exported into the cytoplasm to initiate PPR protein synthesis. (**C**) High-resolution NMR structure (PDB entry 6IZP) of a ^5^UGGAA^9^/^21^UGGAA^25^ pentad (sticks) complexed with two NCD molecules (spheres).

**Figure 4 cells-11-01567-f004:**
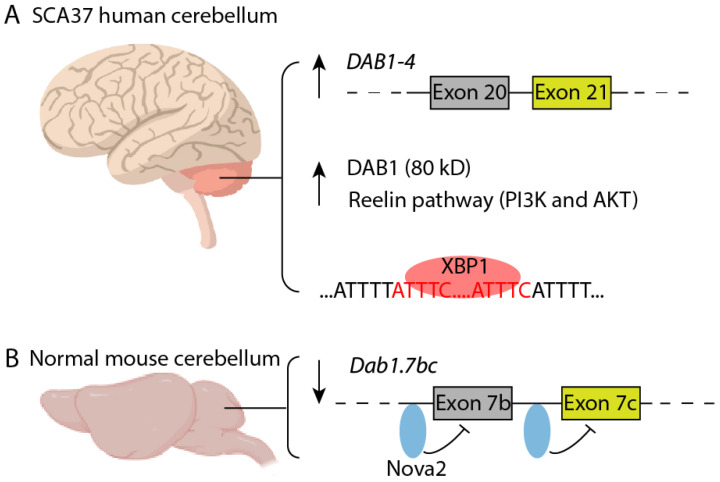
*DAB1* mRNA and protein dysregulation in SCA37 brain. (**A**) *DAB1-4* mRNA (with exons 20 and 21 inclusion) is uniquely expressed in SCA37 but absent in control cerebellum. DAB1 (80 kD) overexpression and reelin pathway overactivation are observed in SCA37 patient cerebellum. In silico prediction identifies ATTTC repeats as binding sites for XBP1. (**B**) In normal mouse cerebellum, the transcription factor Nova2 binds to the YCAY sites upstream of exons 7b and 7c (evolutionarily conserved and color coded as exons 20 and 21) to prevent their inclusion. Overexpression of *Dab1.7bc* mRNA antagonizes WT Dab1 function and causes migration defects of Purkinje cells.

**Table 1 cells-11-01567-t001:** Pentanucleotide expansion diseases with their characteristic pathogenic alleles.

Disease	Gene (Locus)	Pathogenic Repeat Range
SCA10	Intron 9 of *ATXN10* (22q13.31)	ATTCT_800–4500_
SCA31	ntron shared by *BEAN* & *TK2* (16q22.1)	TGGAA_n_
SCA37	5′-UTR intron 1 or 3 of *DAB1* (1p32.2)	ATTTC_31–75_
CANVAS	Intron 2 of *RFC1* (4p14)	AAGGG_400–2000_
		ACAGG_~1000_
		AAAGG_10–25_AAGGG_n_AAAGG_n_

## Data Availability

Not applicable.

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
