# Peer review of "Mechanistic and Therapeutic Insights into Ataxic Disorders with Pentanucleotide Expansions"

_cells, 2022, doi:10.3390/cells11091567_

Round 1

Reviewer 1 Report

This is a very well written, informative and comprehensive review.  I would like to commend the authors on the production.  I have two suggestions for revision:

(1) The last line of the abstract: This review aims to provide new in vitro and in vivo insights, I think this could be amended to this review highlights rather than provides unless novel data have been included?

(2) The area of current therapeutic interventions is only lightly touched upon in the last paragraph.  Could the authors expand that area?

Author Response

This is a very well written, informative and comprehensive review.  I would like to commend the authors on the production.  I have two suggestions for revision:

(1) The last line of the abstract: This review aims to provide new in vitro and in vivo insights, I think this could be amended to this review highlights rather than provides unless novel data have been included?

We thank the reviewer’s comments and have modified the sentence to “This review aims to highlight new in vitro and in vivo insights into these incurable diseases.”

(2) The area of current therapeutic interventions is only lightly touched upon in the last paragraph.  Could the authors expand that area?

As the reviewer suggested, we have expanded the discussion on current therapeutic interventions as follows:

“Our therapeutic toolbox has expanded beyond small molecules to biological modalities, such as DNA- and RNA-editing CRISPRs [122-128], antisense oligonucleotides (ASOs) [129-135], and DNAzymes [136]. For instance, Cas9 proteins complexed with single or dual guide RNAs have been used to excise repeat expansions across multiple neurodegenerative diseases [122-124,128,137,138]. Their relatively large cassette size with respect to adeno-associated viral (AAV) packaging capacity, potential immunogenicity, and off- and on-target deletion and chromosomal rearrangement still pose concerns about the efficacy and safety of permanent DNA editing [122,139,140]. By using Cas-guide mRNAs or ribonucleoproteins encapsulated in lipid nanoparticles (LNPs) for transient expression [141], deactivated Cas9 (dCas9) for blocking repeat transcription [125,142], dCas9 conjugates for repeat RNA degradation [126], or compact Cas proteins (such as CasX, Cas14 or Casφ) [143-145], some of the above obstacles could be circumvented. Alternatively, RNA editing may offer a “safer” approach that would require frequent dosing. For instance, Cas13-based systems have been successfully tested for RNA elimination in DM1 myoblast [127], cancer cells [146,147], and coronavirus-infected cells [148]; it is currently unknown whether the collateral RNA cleavage by Cas13 would outweigh its therapeutic benefits. However, it is reassuring that a Cas13d-based glia-to-neuron conversion significantly alleviated retinal injury in vivo and motor defects in a Parkinson’s mouse model without inducing apparent toxicity [149]. In parallel, both ASO and DNAzyme are RNA-targeting single-stranded DNAs that are compatible with a wide range of chemical modifications; the difference in their modus operandi is that ASO requires cellular RNase H activity, while DNAzyme relies on its own catalytic function induced by ions and 3D folding [150]. Although chemically modified ASOs (and by inference DNAzymes) can already achieve wide central nervous distribution in rodents and non-human primates [151], their packaging with LNP and bio-conjugation with cell-penetrating peptides, receptor-targeting antibodies or ligands may further facilitate the passage across the blood-brain barrier, cellular uptake and endosomal escape.”

Reviewer 2 Report

I would like to start by congratulating the authors on such an interesting paper. The authors provide us a complete review on ataxias due to pentanucleotide expansions, with particular emphasis on the underlying pathogenic mechanisms. This is a fascinating and “hot” topic, with the paper being very well written and pleasant to read.

The different SCAs are not described in a homogenous way. One can understand this, but perhaps they could try to shorten a bit on SCA10, or elaborate more on the remaining ataxias.

I would very much appreciate if the authors could elaborate on the philosophical topic of SCA10, 31 and 37 being AD disorders and CANVAS an AR disorder, in relation with the underlying genetic defects and pathology.

Minor issues:

Page 2 “…gene and characteristic of predominant cerebellar ataxia and epileptic seizure…” this sentence could be rewritten

Page 2 “Extensive brain degeneration in the anterior lobe was also observed in later stages of SCA10 by a magnetic resonance imaging (MRI) study” – what do the authors mean with anterior lobe? Frontal lobe?

Author Response

I would like to start by congratulating the authors on such an interesting paper. The authors provide us a complete review on ataxias due to pentanucleotide expansions, with particular emphasis on the underlying pathogenic mechanisms. This is a fascinating and “hot” topic, with the paper being very well written and pleasant to read.

The different SCAs are not described in a homogenous way. One can understand this, but perhaps they could try to shorten a bit on SCA10, or elaborate more on the remaining ataxias.

We thank the reviewer’s comments and have reduced the volume on SCA10 extensively on pages 2-6.

I would very much appreciate if the authors could elaborate on the philosophical topic of SCA10, 31 and 37 being AD disorders and CANVAS an AR disorder, in relation with the underlying genetic defects and pathology.

We fully agree with the review’s point on differential inheritance patterns and have added the following paragraph on page 12:

“It is currently unclear as to why A/T-rich repeat expansions cause an autosomal dominant inheritance in SCA10, SCA31 and SCA37, but an autosomal recessive inheritance in CANVAS. Given that no apparent changes in mRNA and/or protein expression of the expanded genes were observed between normal and patient samples (of all four discussed diseases), a protein LOF and its related gene dosing effect are unlikely a major contributor to the different inheritance patterns. This is particularly interesting for CANVAS, as Friedreich ataxia, another autosomal recessive neurodegenerative disease caused by A-rich repeat expansions, exhibits a marked reduction in mRNA and protein dosing of frataxin [119,120]. The causative gene of CANVAS – RFC1 – plays an important role in DNA replication and damage repair. It is possible that compensatory mechanisms are in place in individuals with one mutant RFC1 allele, while in individuals with two recessive alleles, ageing and accumulation of DNA damage may eventually overrun the repair mechanism, leading to late-onset neurodegeneration. Indeed, certain mutations in DNA damage repair proteins have been linked to autosomal recessive forms of ataxia and Parkinson’s disease [121]. It is also interesting to note that no sense or antisense RNA foci were observed in CANVAS patient brain, but RFC1 intron 2 retention is consistently elevated in multiple patient brain regions and peripheral tissues [104]. Intron retention typically occurs in G/C-rich, autosomal dominant expansion diseases (such as DM, Fuch’s endothelial corneal dystrophy, and C9orf72), but not in A/T-rich expansion diseases (such as Friedreich ataxia and SCA10); it has a wide cellular impact on tissue development, neuronal gene expression, RNA nuclear retention and nucleocytoplasmic transport, to name but a few [52]. It is currently unknown as to how intron retention in CANVAS links to disease etiology and autosomal recessive inheritance.”

Minor issues:

Page 2 “…gene and characteristic of predominant cerebellar ataxia and epileptic seizure…” this sentence could be rewritten

As the reviewer suggested, we have modified the sentence to: “SCA10, an autosomal dominant neurodegenerative disease, is caused by ATTCT repeat expansion in intron 9 of the ATXN10 gene. The core clinical phenotype is cerebellar ataxia with variably associated epileptic seizure [16].”

Page 2 “Extensive brain degeneration in the anterior lobe was also observed in later stages of SCA10 by a magnetic resonance imaging (MRI) study” – what do the authors mean with anterior lobe? Frontal lobe?

We have clarified the sentence: “Extensive brain degeneration in the anterior cerebellum (lobules I-VI) was also observed in later stages of SCA10 by a magnetic resonance imaging (MRI) study [36].”

Reviewer 3 Report

Authors presented the importance of the ataxias due to the pentanucleotide expansions and its pathomechanisms.

The title needs improvement to relay the correct meaning of the ataxias due to the pentanucleotide expansions. The expression of "pentanucleotide expansion ataxias" is nor appropriate.

Many reports with nucleotide expansion diseases described the ramifications and significance of the expansions with supporting evidences of mRNA and protein profiling, which should be discussed in-depth and will draw much attentions from the readers. 

Author Response

Authors presented the importance of the ataxias due to the pentanucleotide expansions and its pathomechanisms.

The title needs improvement to relay the correct meaning of the ataxias due to the pentanucleotide expansions. The expression of "pentanucleotide expansion ataxias" is nor appropriate.

We thank the reviewer’s comments and have modified the title to: “Mechanistic and therapeutic insights into ataxic disorders with pentanucleotide expansions”.

Many reports with nucleotide expansion diseases described the ramifications and significance of the expansions with supporting evidences of mRNA and protein profiling, which should be discussed in-depth and will draw much attentions from the readers. 

We thank the reviewer’s comments and have added the following paragraph in discussion:

“It is therefore imperative to unbiasedly identify and cross-reference the protein partners of different repeat types. To date, large scale RNA sequencing (RNAseq) and unbiased protein pulldown experiments using animal brains and cell models of SCA31, SCA37 and CANVAS are still missing. We and our collaborators have performed RNAseq analyses on SCA10 patient and mouse brains but, surprisingly, did not find significant splicing alterations in comparison to WT samples (unpublished data).”